# LAYER-WISE ADVERSARIAL DEFENSE: AN ODE PERSPECTIVE

## ABSTRACT

Deep neural networks are observed to be fragile against adversarial attacks, which have dramatically limited their practical applicability. On improving model robustness, the adversarial training techniques have proven effective and gained increasing attention from research communities. Existing adversarial training approaches mainly focus on perturbations to inputs, while the effect of the perturbations in hidden layers remains underexplored. In this work, we propose layer-wise adversarial defense which improves adversarial training by a noticeable margin. The basic idea of our method is to strengthen all of the hidden layers with perturbations that are proportional to the back-propagated gradients. In order to study the layer-wise neural dynamics, we formulate our approach from the perspective of ordinary differential equations (ODEs) and build up its extended relationship with conventional adversarial training methods, which tightens the relationship between neural networks and ODEs. In the implementation, we propose two different training algorithms by discretizing the ODE model with the Lie-Trotter and the Strang-Marchuk splitting schemes from the operator-splitting theory. Experiments on CIFAR-10 and CIFAR-100 benchmarks show that our methods consistently improve adversarial model robustness on top of widely-used strong adversarial training techniques.

## 1 INTRODUCTION

Recent years have witnessed the prosperity of deep learning in many tasks (Hinton & Salakhutdinov, 2006; Sutskever et al., 2014; He et al., 2016; LeCun et al., 2015; Huang et al., 2017; Vaswani et al., 2017). Stacked with multiple layers, neural networks provide an end-to-end solution to all the tasks and prove to be highly effective. However, the seminal study by Szegedy et al. (2013) has shown that deep neural networks (DNNs) can be fragile against attacks: minor perturbations on inputs lead to significant change in model predictions. Regarding the defense approaches, intensive studies on adversarial defense techniques have been proposed (Athalye et al., 2018a; Goodfellow et al., 2014; Zheng et al., 2016; Madry et al., 2018; Zhang et al., 2019b; Kurakin et al., 2017; Pang et al., 2019a; 2020; 2019b; Raff et al., 2019; Guo et al., 2018; Zhang et al., 2020a; Balunovic & Vechev, 2019; Wong et al., 2020; Chan et al., 2020; Zhang et al., 2020b). Among these techniques, adversarial training algorithms (Madry et al., 2018; Zhang et al., 2019b) incorporate the effect of perturbed inputs into the loss function, which are shown to be competent and boasts the dominant impact in the adversarial defense research field.

While adversarial training techniques have gained increasing attention in the robust deep learning research community, most of current approaches concentrate on deriving perturbations on the inputs with gradients back-propagated from the loss function. However, as information flow in neural networks starts from inputs and passes through hidden layers, it is essential to robustify both the inputs and the hidden layers. While previous studies have made successful attempts on introducing damping terms (Yang et al., 2020) or stochastic noise (Liu et al., 2020; Wang et al., 2019) to each layer in neural architectures, they concentrate on improving general model robustness and are less focused on adversarial model robustness. We ask the following question: *Can we take the hidden layers of neural networks into account to improve adversarial model robustness?*

In this work, we propose layer-wise adversarial defense to improve adversarial training, which enhances adversarial model robustness by stabilizing both inputs and hidden layers. In our method,

the layer-wise perturbations are incorporated into the robust optimization framework of adversarial training. We propose to inject scaled back-propagated gradients into the architecture as layer-wise perturbations. Besides, we formulate our method from the perspective of ordinary differential equations and propose a novel ODE as its the continuous limit in order to study the neural dynamics. Inspired from the rich literature on numerical analysis, we use the Lie-Trotter and the Strang-Marchuk splitting schemes to solve the proposed ODE. We refer to the resulted discrete algorithms as Layer-wise Adversarial Defense (LAD) and LAD-SM, respectively. Furthermore, we build up the extended relationship between our methods with current natural training and adversarial training techniques by analyzing the second order dynamics. Our analysis shows that our methods have introduced additional perturbations in the first order initial value of the second order dynamics compared with current adversarial training algorithms.Experiments on the CIFAR-10 and CIFAR-100 benchmarks show that our methods improve adversarial model robustness on top of different widely-used strong adversarial training techniques.

We summarize our contributions as follows:

- We propose layer-wise adversarial defense which generalizes conventional adversarial training approaches with layer-wise adversarial perturbations (Section 3.1);

- We investigate the continuous limit of our layer-wise adversarial defense methods and propose an ODE that integrates the adjoint state into the forward dynamics (Section 3.2);

- We build up the extended relationship between our methods and current adversarial training approaches by analyzing the second order neural dynamics in theory. Experiments have also shown the effectiveness of our methods in practice. (Section 3.3 and Section 4).

## 2 RELATED WORK

### 2.1 ADVERSARIAL MODEL ROBUSTNESS

In this section we review the literature on gradient-based attack and defense approaches in the field of adversarial model robustness. For adversarial attacks, widely-used approaches include Fast Gradient Sign Method (FGSM) (Goodfellow et al., 2015) and Iterated Fast Gradient Sign Method (IFGSM) (Madry et al., 2018). For a given data point, FGSM induces the adversarial example by moving with the attack radius $\epsilon$ at each component along the gradient ascent direction. Iterated FGSM performs FGSM with inner iteration updates with smaller step size $\alpha$. Prior studies have inspired multiple adversarial attack techniques (Athalye et al., 2018b; Carlini & Wagner, 2017; Ilyas et al., 2018; Dong et al., 2018; Pang et al., 2018). Adversarial defense techniques can be categorized by training phase (Athalye et al., 2018a; Goodfellow et al., 2014; Zheng et al., 2016; Madry et al., 2018; Zhang et al., 2019b; Kurakin et al., 2017; Pang et al., 2019a; 2020; Zhang et al., 2020a; Balunovic & Vechev, 2019; Wong et al., 2020; Chan et al., 2020; Zhang et al., 2020b) and inference phase (Pang et al., 2019b; Raff et al., 2019; Xie et al., 2018; Guo et al., 2018).

The widely-used approach in training phase is Projected Gradient Descent (PGD) training (Madry et al., 2018), which integrates the effect of the perturbed inputs into its loss function. The current state-of-the-art defense approach in training phase is TRADES (Zhang et al., 2019b), which additionally introduces the boundary error as a regularization term into its loss function. In our experiments, we select PGD training and TRADES as our baselines. While substantially enhancing adversarial model robustness, the gradient-based perturbations in adversarial training are currently only performed on inputs. As cascaded hidden layers comprise the passage for information flow in neural networks, it is essential to stabilize hidden layers as well. In our work, we introduce layer-wise gradient-based perturbations to neural architectures to improve adversarial model robustness.

### 2.2 ODE-INSPIRED ARCHITECTURE DESIGNS

Research about the relationship between neural networks and ODEs starts with the continuous limit formulation of ResNet (E, 2017), which has inspired many novel neural architecture designs (Lu et al., 2018; Zhu et al., 2018; Chang et al., 2018; Haber & Ruthotto, 2017; Chen et al., 2018; Dupont et al., 2019). Regarding model robustness, most prior studies have focused on improving dynamic system stability by Lyapunov analysis, more stable numerical schemes, or imposing regularization.

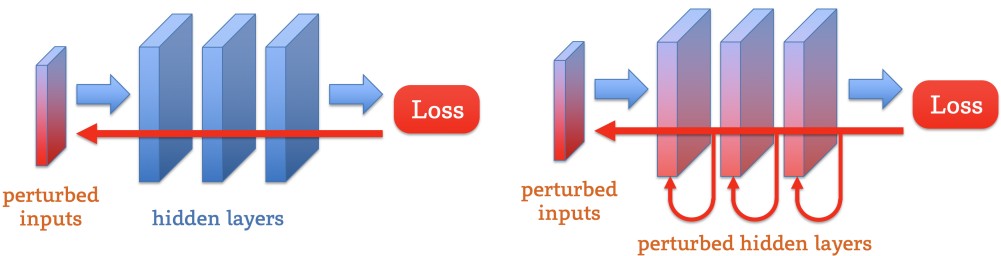

Figure 1: Left: Conventional adversarial training techniques. Right: Our layer-wise adversarial defense methods. In conventional adversarial training algorithms, the perturbations are only performed on the input side with gradients back-propagated through the architecture. In our layer-wise adversarial defense methods, the gradients on each layers are recorded along backpropagation, which are further added to each layers as layer-wise perturbations in another forward pass.

From Lyaponov stability perspective, Yang et al. (2020) introduce damping terms to residual networks to stabilize dynamic systems. Appropriately adjusting the damping factor introduces damping effect to the dynamic system and enhances model robustness. Similarly, Chang et al. (2019) improve Lyapunov stability of RNNs by imposing antisymmetric constraints on the weight matrix. On more stable numerical schemes, prior studies include taking small step sizes in the forward Euler scheme (Zhang et al., 2019c) or leveraging implicit schemes to enhance stability (Reshniak & Webster, 2019; Li et al., 2020). For imposing regularization, stochasticity is introduced into ODEs for stabilization (Liu et al., 2020; Wang et al., 2019). Hanshu et al. (2019) regularize neural ODE models to be time-invariant and add a regularization term about the upper bound of the effect from input perturbation. Zhang et al. (2019a) propose a differential game formulation for adversarial training and accelerate the process from the optimal control theory. Our work differs from the prior studies by integrating gradient-based perturbations into the neural dynamics, which proves to be an extension to current approaches on improving adversarial model robustness in both theory and practice.

## 3 LAYER-WISE ADVERSARIAL DEFENSE

### 3.1 THE MODEL FORMULATION

The objective of conventional adversarial training approaches can be formulated into a min-max problem (Madry et al., 2018; Zhang et al., 2019a). We introduce layer-wise perturbations $\{\Delta \mathbf{x}_n\}_{n=0}^N$ and rewrite the min-max problem as follows:

$$\min_{\{\boldsymbol{\theta}_n\}_{n=1}^N} \max_{\{\Delta \mathbf{x}_n\}_{n=0}^N} L(\boldsymbol{\Theta}, \Delta \mathbf{x}) := L(\mathbf{x}_N)$$

$$\text{subject to} \quad \tilde{\mathbf{x}}_0 = \mathbf{x}_0 + \Delta \mathbf{x}_0,$$
$$\mathbf{x}_{n+1} = \tilde{\mathbf{x}}_n + f(\tilde{\mathbf{x}}_n, \boldsymbol{\theta}_n), \tag{1}$$
$$\tilde{\mathbf{x}}_{n+1} = \mathbf{x}_{n+1} + \Delta \mathbf{x}_{n+1}, \ n = 0, 1, 2, \cdots, N-1.$$

where $N$ is the number of layers in a neural network, $\boldsymbol{\Theta} = \{\boldsymbol{\theta}_n\}_{n=1}^N$ represent its trainable parameters and the $\{\Delta \mathbf{x}_n\}_{n=0}^N$ represent the layer-wise perturbations. In our formulation, we ignore the bounded assumptions on the perturbations for simplicity, since if there are additional bounded constraints on the perturbations, we can project the gradient onto the intervals. It is noted that when $\Delta \mathbf{x}_n = 0$ for all $n = 1, \ldots, N$, the model (1) reduces to the conventional adversarial training formulation. More specifically, for adversarial training algorithms (Madry et al., 2018; Zhang et al., 2019b), let $M$ be the maximum number of inner iterations for the perturbations, we have the following update rule (ignoring the bounded constraints):

$$\mathbf{x}_0^{(m+1)} = \mathbf{x}_0^{(m)} + \eta \left. \frac{\partial L}{\partial \mathbf{x}_0} \right|_{(m)}, \ \mathbf{x}_0^{(0)} = \mathbf{x}_0, \tag{2}$$

where the perturbation $\Delta \mathbf{x}_0$ is defined as $\mathbf{x}_0^{(M)} - \mathbf{x}_0$.

In this work, we generalize the above idea to introducing perturbations to hidden layers, i.e. determine $\Delta \mathbf{x}_{n+1}$ in Eq. (1) so that the objective in Eq. (1) is maximized. Similar to Eq. (2), we perturb

$\mathbf{x}_{n+1}$ with iterative gradient descent:

$$\mathbf{x}_{n+1}^{(m+1)} = \mathbf{x}_{n+1}^{(m)} + \eta \frac{\partial L}{\partial \mathbf{x}_{n+1}} \Big|_{(m)}, \ \mathbf{x}_{n+1}^{(0)} = \mathbf{x}_{n+1}, \tag{3}$$

where $\eta$ is the step size as a scaling factor over the gradients $\partial L / \partial \mathbf{x}_{n+1}$. We set $\Delta \mathbf{x}_{n+1} = \mathbf{x}_{n+1}^{(M)} - \mathbf{x}_{n+1}$, where $M$ is the number of the iteration steps. When $M = 1$, the layer-wise adversarial perturbations are given by

$$\Delta \mathbf{x}_{n+1} = \mathbf{x}_{n+1}^{(1)} - \mathbf{x}_{n+1} = \eta \frac{\partial L}{\partial \mathbf{x}_{n+1}}, \ n = 0, 1, 2, \cdots, N-1. \tag{4}$$

Replacing the maximization of layer-wise perturbations $\{\Delta \mathbf{x}_n\}_{n=0}^N$ by Eq. (4), we obtained a simplified problem as follows.

$$\min_{\{\boldsymbol{\theta}_n\}_{n=1}^N} L(\boldsymbol{\Theta}) := L(\mathbf{x}_N)$$
$$\text{subject to} \quad \tilde{\mathbf{x}}_{n+1} = \mathbf{x}_n + f(\mathbf{x}_n, \boldsymbol{\theta}_n), \tag{5}$$
$$\mathbf{x}_{n+1} = \tilde{\mathbf{x}}_{n+1} + \eta \frac{\partial L}{\partial \tilde{\mathbf{x}}_{n+1}}, \ n = 0, 1, 2, \cdots, N-1,$$

with the inputs $\tilde{\mathbf{x}}_0 = \mathbf{x}_0 + \Delta \mathbf{x}_0$ determined by Eq. (4). Notice that directly applying Eq. (5) requires alternative computations of $\tilde{\mathbf{x}}_{n+1}$ and $\partial L / \partial \tilde{\mathbf{x}}_{n+1}$ with iterative forward and backward passes, which can be extremely time-consuming. In our implementation, we leverage a two-stage approach: first record the gradients with respect to each layer in a forward and backward pass, then add the recorded gradients to each layer in another forward pass as layer-wise adversarial perturbations. We refer to this algorithm as our Layer-wise Adversarial Defense (LAD) method.

## 3.2 ODE-Inspired Algorithm Design

In this section, we explore the continuous formulation of the constraints in Eq. (5) to study the layer-wise dynamics from the ODE perspective. Recall that the conventional ODE formulation (E, 2017) takes the following form:

$$\frac{d\hat{\mathbf{x}}(t)}{dt} = f(\hat{\mathbf{x}}(t), t), \ \hat{\mathbf{x}}(0) = \hat{\mathbf{x}}_0, \tag{6}$$

which is the continuous limit of the discrete ResNet $\hat{\mathbf{x}}_{n+1} = \hat{\mathbf{x}}_n + f_n(\hat{\mathbf{x}}_n)$ with the time step $\Delta t = 1$. In the conventional ODE, $L = L(\hat{\mathbf{x}}(T))$ is defined as the loss function. In our work, we propose to integrate $dL/d\hat{\mathbf{x}}(t)$ into the forward process:

$$\frac{d\mathbf{x}(t)}{dt} = f(\mathbf{x}(t), t) + \eta \frac{dL}{d\hat{\mathbf{x}}(t)}, \ \mathbf{x}(0) = \mathbf{x}_0. \tag{7}$$

where $\mathbf{x}_0$ is the (perturbed) inputs and $\eta$ is a scaling factor. The introduced $dL/d\hat{\mathbf{x}}(t)$ represents the continuous limit of back-propagated gradients. As we record the gradients from an earlier backward pass, the corresponding forward pass can be approximately treated as solving the original ODE. As shown in Eq. (7), there are two operators on the right hand side. According to the rich literature on the operator-splitting theory from numerical analysis, we have the following proposition.

**Proposition 3.1.** *The LAD method is the numerical solution of our proposed ODE with the Lie-Trotter splitting scheme with step size $\Delta t = 1$.*

In the operator splitting theory, the Strang-Marchuk (SM) splitting scheme (Ascher & Petzold, 1998) is also a widely-used technique of solving ODE with multiple terms. Compared with the Lie-Trotter splitting scheme, the Strang-Marchuk splitting scheme enjoys much lower local truncation errors (Bobylev & Ohwada, 2001). Lu et al. (2019) propose to leverage the SM splitting scheme to improve the Transformer architecture, which results in higher model accuracy. We also propose to use the SM splitting scheme to discretize our ODE in Eq. (7), but the direct application of SM method is intractable. With proper approximation, we have the following modified version.

**Theorem 3.2.** *An approximated numerical scheme of Eq. (7) with the SM splitting is*

$$\begin{cases} \tilde{\mathbf{x}}_n = \mathbf{x}_n + \frac{\eta}{2} \frac{\partial L}{\partial \hat{\mathbf{x}}_n}, \\ \bar{\mathbf{x}}_n = \tilde{\mathbf{x}}_n + f_n(\tilde{\mathbf{x}}_n), \\ \mathbf{x}_{n+1} = \bar{\mathbf{x}}_n + \frac{\eta}{4} \left( \frac{\partial L}{\partial \hat{\mathbf{x}}_n} + \frac{\partial L}{\partial \tilde{\mathbf{x}}_{n+1}} \right). \end{cases} \tag{8}$$

The proofs and a self-contained introduction to the numerical background are provided in Appendix A. We refer to Eq. (8) as our LAD-SM method.

## 3.3 ANALYSIS OF THE SECOND ORDER DYNAMICS

In this section, we provide an analysis of the second order dynamics and connect our ODE (Eq. (7)) with the original ODE (Eq. (6)). Define $\mathbf{a}(t) = \mathrm{d}L/\mathrm{d}\hat{\mathbf{x}}(t)$, it is known that the dynamics of $\mathbf{a}(t)$ (Pontryagin et al., 1962; Chen et al., 2018) satisfies

$$\frac{\mathrm{d}\mathbf{a}}{\mathrm{d}t} = -\mathbf{a}(t)^{\mathrm{T}}\frac{\partial f(\hat{\mathbf{x}}, t)}{\partial \hat{\mathbf{x}}}. \tag{9}$$

The next theorem presents the second order dynamics of the proposed Eq. (7).

**Theorem 3.3.** *The second order dynamics of Eq. (7) is given by*

$$\frac{\mathrm{d}^2\mathbf{x}}{\mathrm{d}t^2} = \frac{\partial f(\mathbf{x}, t)}{\partial \mathbf{x}}f(\mathbf{x}, t) + \frac{\partial f(\mathbf{x}, t)}{\partial t} + \eta\mathbf{a}(t)^{\mathrm{T}}\left(\frac{\partial f(\mathbf{x}, t)}{\partial \mathbf{x}} - \frac{\partial f(\hat{\mathbf{x}}, t)}{\partial \hat{\mathbf{x}}}\right) \tag{10}$$

*with*

$$\mathbf{x}(0) = \mathbf{x}_0, \quad \frac{\mathrm{d}\mathbf{x}}{\mathrm{d}t}\bigg|_{t=0} = f(\mathbf{x}(0), 0) + \eta\frac{\mathrm{d}L}{\mathrm{d}\mathbf{x}(0)}. \tag{11}$$

*Proof.* By the direct computation, we know the second order dynamics of Eq. (7) is

$$\frac{\mathrm{d}^2\mathbf{x}}{\mathrm{d}t^2} = \frac{\partial f(\mathbf{x}, t)}{\partial \mathbf{x}}\frac{\mathrm{d}\mathbf{x}}{\mathrm{d}t} + \frac{\partial f(\mathbf{x}, t)}{\partial t} - \eta\mathbf{a}(t)^{\mathrm{T}}\frac{\partial f(\hat{\mathbf{x}}, t)}{\partial \hat{\mathbf{x}}} \tag{12}$$

$$= \frac{\partial f(\mathbf{x}, t)}{\partial \mathbf{x}}\Big(f(\mathbf{x}, t) + \eta\mathbf{a}(t)\Big) + \frac{\partial f(\mathbf{x}, t)}{\partial t} - \eta\mathbf{a}(t)^{\mathrm{T}}\frac{\partial f(\hat{\mathbf{x}}, t)}{\partial \hat{\mathbf{x}}} \tag{13}$$

$$= \frac{\partial f(\mathbf{x}, t)}{\partial \mathbf{x}}f(\mathbf{x}, t) + \frac{\partial f(\mathbf{x}, t)}{\partial t} + \eta\mathbf{a}(t)^{\mathrm{T}}\left(\frac{\partial f(\mathbf{x}, t)}{\partial \mathbf{x}} - \frac{\partial f(\hat{\mathbf{x}}, t)}{\partial \hat{\mathbf{x}}}\right) \tag{14}$$

where the first equality is from Eq. (9) and the second equlaity is from Eq. (7). $\square$

Moreover, the second order dynamics of Eq. (6) is given by

$$\frac{\mathrm{d}^2\hat{\mathbf{x}}}{\mathrm{d}t^2} = \frac{\partial f(\hat{\mathbf{x}}, t)}{\partial \hat{\mathbf{x}}}f(\hat{\mathbf{x}}, t) + \frac{\partial f(\hat{\mathbf{x}}, t)}{\partial t},$$

$$\text{with} \quad \hat{\mathbf{x}}(0) = \mathbf{x}_0, \quad \frac{\mathrm{d}\hat{\mathbf{x}}}{\mathrm{d}t}\bigg|_{t=0} = f(\mathbf{x}_0, 0). \tag{15}$$

Since $\mathbf{x}$ is a small perturbation of $\hat{\mathbf{x}}$ as well as the existence of $\eta\mathbf{a}(t)$, the last term in Eq. (10) can be negligible and the main difference of the seccond order dynamics of our ODE (Eq. (7)) with the original ODE (Eq. (6)) lies in the first order initial values. The extra momentum of the input in Eq. (11) leads to extra perturbations in all of the hidden layers during the propagation process. The implementation details of our methods can be found in Appendix B.

## 4 EXPERIMENTS

We evaluate our proposed methods on the CIFAR (Krizhevsky et al., 2009) benchmarks. For each experiment, we conduct 3 runs with different random seeds and report the averaged result to reduce the impact of random variations. We also select Yang et al. (2020) as our baseline methods, which introduce layer-wise damping terms to each layer in neural networks. The details of experimental settings can be found in Appendix B.

## 4.1 LAD with Natural Training

We first evaluate our methods in the setting of natural training (He et al., 2016). This setting is equivalent to setting $\Delta\mathbf{x}_0 = 0$ in our layer-wise adversarial defense framework (Eq. (1)). Table 4.1 shows the accuracy and robustness results of our methods composed with natural training under different attack radii $\epsilon$.

Table 1: Accuracy and robustness results of natural training baselines as well as our methods on CIFAR-10 and CIFAR-100. The "Avg" column reports the averaged robustness results for each methods under different attacks of all the radii. "Natural" denotes ResNet with natural training; "damp" and "$\lambda$-damp" represent In-ResNet and $\lambda$-In-ResNet in Yang et al. (2020), respectively.

| Benchmark | Method | Acc | FGSM | | IFGSM | | Avg |
|---|---|---|---|---|---|---|---|
| | | | 0.5/255 | 1/255 | 0.5/255 | 1/255 | |
| CIFAR-10 | natural | 92.89 | 67.22 | 49.80 | 56.98 | 19.20 | 48.30 |
| | natural + damp | 92.77 | 70.91 | **53.17** | 64.26 | 27.95 | 54.07 |
| | natural + $\lambda$-damp | 92.90 | 70.75 | 52.70 | 63.73 | 26.68 | 53.47 |
| | natural + LAD | 93.04 | **71.79** | 51.18 | **66.02** | **28.27** | **54.32** |
| | natural + LAD-SM | **93.05** | 70.91 | 50.82 | 65.20 | 27.36 | **53.57** |
| CIFAR-100 | natural | **71.35** | 33.27 | 21.25 | 24.15 | 5.54 | 21.05 |
| | natural + damp | 70.89 | 35.75 | 22.10 | 28.14 | 7.35 | 23.34 |
| | natural + $\lambda$-damp | 70.44 | 35.83 | 22.13 | 27.94 | 7.42 | 23.33 |
| | natural + LAD | 70.21 | 39.74 | 22.95 | 33.81 | 9.90 | **26.60** |
| | natural + LAD-SM | 70.28 | **40.33** | **23.92** | **34.74** | **11.30** | **27.57** |

Results show that our methods perform best under most attack settings with small radii. While all methods fail under attacks with larger radii because of the vulnerability of natural training, results under attacks with small radii have still shown the enhanced robustness by only introducing perturbations on hidden layers.

## 4.2 LAD with PGD Training

In this section, we report our experiments with PGD Training (Madry et al., 2018). Table 2 shows the results of our methods and the baseline techniques with PGD training.

Table 2: Accuracy and robustness results of baselines as well as our methods with PGD training on the CIFAR-10 and CIFAR-100 benchmarks. The "Avg" column reports the averaged robustness results for each methods under different attacks of all the radii. "PGD + damp" and "PGD + $\lambda$-damp" represent In-ResNet and $\lambda$-In-ResNet in Yang et al. (2020) with adversarial training, respectively.

| Benchmark | Method | Acc | FGSM | | IFGSM | | Avg |
|---|---|---|---|---|---|---|---|
| | | | 8/255 | 12/255 | 8/255 | 12/255 | |
| CIFAR-10 | PGD | **82.14** | 53.55 | 42.48 | 44.79 | 25.46 | 41.57 |
| | PGD + damp | 82.12 | 53.39 | 41.96 | 44.43 | 25.31 | 41.27 |
| | PGD + $\lambda$-damp | 82.58 | 53.89 | 42.40 | 44.51 | 25.41 | 41.55 |
| | PGD + LAD | 82.03 | **53.99** | 42.37 | 46.07 | 26.88 | **42.33** |
| | PGD + LAD-SM | 81.67 | 53.79 | **42.60** | **46.21** | **27.45** | **42.51** |
| CIFAR-100 | PGD | 56.85 | 23.71 | 16.10 | 17.74 | 8.52 | 16.52 |
| | PGD + damp | 53.16 | 19.89 | 13.22 | 12.40 | 5.22 | 12.68 |
| | PGD + $\lambda$-damp | 56.14 | 19.27 | 13.06 | 12.03 | 5.51 | 12.47 |
| | PGD + LAD | **58.11** | 26.70 | **17.99** | **22.44** | **11.85** | **19.75** |
| | PGD + LAD-SM | 57.97 | **26.79** | 17.94 | 22.39 | 11.58 | **19.68** |

According to the experimental results, it is shown that our methods consistently outperform the baselines in the robustness performance. Another interesting finding is that in our experiments, our approaches achieve more significant improvements over baselines on the CIFAR-100 benchmark. We provide a possible interpretation: from the discrete neural network perspective, the stacked

layers represent different levels of representation learning. While our methods have perturbed each layers, it can be interpreted as an augmentation with adversarial "examples" in each levels of feature learning. Given that each class in the CIFAR-100 training set consists only 500 images (which is far less than that in the CIFAR-10 training set (5,000 images)), it is inferred that our methods have a potential positive effect in the data-scarce regime.

## 4.3 LAD WITH TRADES

In this section, we report our experimental results with the TRADES adversarial training approach (Zhang et al., 2019b). Table 3 shows the results of our methods composed with the TRADES method. According to the results shown in Table 2, we only select TRADES with the original ResNet architecture as our baseline methods. Results show that our methods have also surpassed the baseline methods on top of the state-of-the-art TRADES technique. We also provide robustness results of our methods as well as the baselines under stronger attacks in Appendix C.

Table 3: Accuracy and robustness results of the baselines as well as our methods with TRADES defense on the CIFAR-10 and CIFAR-100 benchmarks. The "Avg" column reports the averaged robustness results for each methods under different attacks of all the radii.

| Benchmark | Method | Acc | FGSM | | IFGSM | | Avg |
|---|---|---|---|---|---|---|---|
| | | | 8/255 | 12/255 | 8/255 | 12/255 | |
| CIFAR-10 | TRADES | **79.08** | 54.70 | 45.09 | 49.85 | 34.69 | 46.08 |
| | TRADES + LAD | 78.47 | **55.19** | **45.53** | **50.80** | **35.86** | **46.85** |
| | TRADES + LAD-SM | 78.49 | 55.13 | 45.26 | 50.68 | 35.60 | **46.67** |
| CIFAR-100 | TRADES | 52.53 | 28.51 | 21.38 | 25.86 | 16.87 | 23.16 |
| | TRADES + LAD | 52.46 | **29.30** | 21.70 | **27.10** | **17.83** | **23.98** |
| | TRADES + LAD-SM | **52.60** | 29.16 | **21.78** | 26.97 | 17.78 | **23.92** |

## 4.4 LAD WITH STOCHASTICITY

Prior studies have shown that model robustness is improved with layer-wise noise injection as regularization (Liu et al., 2020; Wang et al., 2019). We also experiment on introducing stochasticity to our methods in order to further boost the robustness performance. We augment our experiments with PGD training with additive Gaussian noise, which is proposed by Wang et al. (2019). During training, Gaussian noise $n \sim N(0, 0.01)$ is added to each layer in the model. During inference, we perform forward pass with noise sampled for 100 times and take the averaged output logits as the expectation outputs. Table 4 shows the results on the CIFAR-10 benchmark. As shown from the results, our approaches consistently surpass the baseline approaches under both the deterministic and the stochastic settings.

Table 4: Comparison of accuracy and robustness results of baselines and our methods with PGD training between the deterministic and the stochastic versions on CIFAR-10. The "Avg" column reports the averaged robustness results for each methods under different attacks of all the radii.

| Method | Acc | FGSM | | IFGSM | | Avg |
|---|---|---|---|---|---|---|
| | | 8/255 | 12/255 | 8/255 | 12/255 | |
| PGD | **82.14** | 53.55 | 42.48 | 44.79 | 25.46 | 41.57 |
| PGD + noise | **82.48** | 53.77 | 42.54 | 44.63 | 25.51 | 41.61 |
| PGD + LAD | 82.03 | **53.99** | 42.37 | 46.07 | 26.88 | **42.33** |
| PGD + LAD-SM | 81.67 | 53.79 | **42.60** | 46.21 | 27.45 | 42.51 |
| PGD + LAD + noise | 81.79 | 54.08 | 42.41 | 46.34 | 27.16 | **42.50** |
| PGD + LAD-SM + noise | 81.99 | **54.35** | **43.00** | **46.49** | **27.54** | **42.85** |

### 4.5 THE EFFECT OF $\eta$

In this section, we introduce how we determine the scaling factor $\eta$. The hyperparameter $\eta$ scales the back-propagated gradients, which are further added to the network layer-wisely. We start to get basic knowledge about the proper range of $\eta$ by comparing the norms of back-propagated gradients with the norms of each layers. According to Eq. (11), our methods have essentially perturbed the first order initial value in the view of the second order dynamics. As a result, we let

$$\text{ratio} = \frac{\|\mathbf{x}(0) + f(\mathbf{x}(0), 0)\|_2}{\left\|\frac{\mathrm{d}L}{\mathrm{d}\mathbf{x}(0)}\right\|_2} \tag{16}$$

and trace the ratio during training.

As Figure 2 shows, the ratio is ultimately large at the beginning of training (around $2 \times 10^6$) and becomes smaller during training. While it achieves its minimum at around the end of the training, the ratio is always larger than $1 \times 10^5$. As a result, the ratio serves as a quantitative comparison between the norm of the back-propagated gradient with the norm of the layer in a network. It suggests that the $\eta$ should not be too small; otherwise the layer-wise perturbations can be too small to affect the training. Model robustness results under different $\eta$'s have also proven the finding. As Table 5 shows, the effect on adversarial model robustness from a small $\eta$ is marginal compared with that from a larger $\eta$. Table 5 also shows that $\eta$ should not be too large. A too large $\eta$ may unstabilize the training process and deteriorate the model robustness.

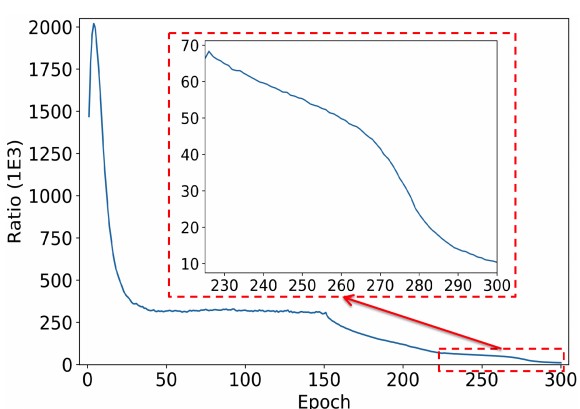

Figure 2: The change of the ratio (Eq. (16)) during PGD training on the CIFAR-100 benchmark.

Table 5: Robustness results under FGSM and IFGSM attacks with radius $= 8/255$ on the CIFAR-100 benchmarks. Our methods are equivalent to the ResNet baseline when $\eta = 0$. It can be seen that $\eta$ should be neither too small nor too large. When $\eta$ grows larger in a proper range, the adversarial model robustness is gradually improved.

| Attack | Method | $\eta$ | | | | | |
|--------|--------|---|---|----|-----|-------|--------|
| | | 0 | 1 | 10 | 100 | 1,000 | 10,000 |
| FGSM | LAD | 23.71 | 23.02 | 24.97 | 23.09 | **26.70** | 17.07 |
| | LAD-SM | | 24.49 | 22.43 | 24.31 | **26.79** | 17.87 |
| IFGSM | LAD | 17.74 | 17.42 | 19.59 | 18.01 | **22.44** | 15.20 |
| | LAD-SM | | 18.97 | 15.98 | 19.54 | **22.39** | 15.30 |

## 5 CONCLUSION

In this work, we propose layer-wise adversarial defense as an extension to current adversarial training approaches. The hidden layers are robustified by the introduced layer-wise perturbations, which are proportional to the back-propagated gradients. We build up the extended relationship of our methods with conventional adversarial training methods from the ODE perspective by providing the analysis of the second order dynamics. We use the Lie-Trotter and the Strang-Marchuk splitting schemes to discretize the proposed ODE model, resulting in two different training algorithms. Experiments on the CIFAR-10 and CIFAR-100 benchmarks show that our methods improve adversarial model robustness on top of different widely-used strong adversarial training techniques.

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

APPENDIX

## A  A BRIEF INTRODUCTION TO THE SPLITTING SCHEMES

### A.1  NUMERICAL BACKGROUND

Consider an ODE with two coupled terms in the right-hand side:

$$\frac{\mathrm{d}\mathbf{x}}{\mathrm{d}t} = F(\mathbf{x}(t), t) + G(\mathbf{x}(t), t), \ \mathbf{x}(0) = \mathbf{x}_0. \tag{17}$$

It is difficult to solve the ODEs with two coupled terms in the right-hand side. The splitting methods provide a natural way to decompose the coupled terms into individual calculation for different differential operators (McLachlan & Quispel, 2002). The simplest splitting scheme is the Lie-Trotter splitting scheme which alternatively calculates $F(\cdot)$ and $G(\cdot)$. The Lie-Trotter splitting scheme with the forward Euler method discretizes Eq. (17) as follows:

$$\begin{aligned}
\tilde{\mathbf{x}}(t) &= \mathbf{x}(t) + \Delta t F(\mathbf{x}(t), t), \\
\mathbf{x}(t + \Delta t) &= \tilde{\mathbf{x}}(t) + \Delta t G(\mathbf{x}(t), t).
\end{aligned} \tag{18}$$

The Lie-Trotter splitting scheme first solves the ODE with respect to $F(\cdot)$ to acquire the intermediate state $\tilde{\mathbf{x}}(t)$. Starting from $\tilde{\mathbf{x}}(t)$, it continues to solve the ODE with respect to $G(\cdot)$ to complete the discretization from time $t$ to time $t + \Delta t$. In the proposed ODE (Eq. (7)), we treat the $f$ function as the operator $F(\cdot)$ and the $\mathrm{d}L/\mathrm{d}\mathbf{x}$ function as the operator $G(\cdot)$. From the formulation of the Lie-Trotter splitting scheme (Eq. (18)), we have the following discretization:

$$\begin{cases}
\tilde{\mathbf{x}}_{n+1} = \mathbf{x}_n + f_n(\mathbf{x}_n), \\
\mathbf{x}_{n+1} = \tilde{\mathbf{x}}_{n+1} + \eta \frac{\partial L}{\partial \tilde{\mathbf{x}}_{n+1}},
\end{cases} \tag{19}$$

which is equivalent to our two-stage approximation approach (note that since we use the first order forward Euler method for each ODE, it is equivalent to either use $\partial L/\partial \hat{\mathbf{x}}_{n+1}$ or use $\partial L/\partial \hat{\mathbf{x}}_n$ in solving $G(\cdot)$). It is thus straightforward to see that Proposition 3.1 holds.

The Strang-Marchuk splitting scheme extends the Lie-Trotter splitting scheme by dividing the one-step solver for $G(\cdot)$ into two half steps. Using the Strang-Marchuk splitting scheme to solve Eq. (17) yields

$$\begin{aligned}
\hat{\mathbf{x}}(t) &= \mathbf{x}(t) + \frac{\Delta t}{2} G(\mathbf{x}(t), t), \\
\tilde{\mathbf{x}}(t) &= \hat{\mathbf{x}}(t) + \Delta t F(\hat{\mathbf{x}}(t), t), \\
\mathbf{x}(t + \Delta t) &= \tilde{\mathbf{x}}(t) + \frac{\Delta t}{2} G\left(\tilde{\mathbf{x}}(t), t + \frac{\Delta t}{2}\right).
\end{aligned} \tag{20}$$

The Strang-Marchuk splitting scheme enjoys lower local truncation error than the Lie-Trotter splitting scheme (Bobylev & Ohwada, 2001). As a result, the Strang-Marchuk splitting scheme may lead to a more accurate solution in terms of solving the proposed ODE (Eq. (7)). In the next section we provide the proof for Theorem 3.2.

### A.2  PROOF OF THEOREM 3.2

Applying the Strang-Marchuk splitting scheme (Eq. (20)) with step size $\Delta t = 1$ to solve our ODE (7), we have the following algorithm:

$$\begin{cases}
\tilde{\mathbf{x}}_n = \mathbf{x}_n + \frac{\eta}{2} \frac{\partial L}{\partial \tilde{\mathbf{x}}_n}, \\
\bar{\mathbf{x}}_n = \tilde{\mathbf{x}}_n + f_n(\tilde{\mathbf{x}}_n), \\
\mathbf{x}_{n+1} = \bar{\mathbf{x}}_n + \frac{\eta}{2} \frac{\partial L}{\partial \tilde{\mathbf{x}}(n+1/2)}.
\end{cases} \tag{21}$$

In Equation (21), the essential part is to calculate $\partial L/\partial \hat{\mathbf{x}}(n+1/2)$. As shown in Eq. (9), our goal is to estimate $\mathbf{a}(n+1/2)$ with $\mathbf{a}(n)$ and $\mathbf{a}(n+1)$. As the learned filters in the deep layers converges, we can mildly treat $\partial f(\hat{\mathbf{x}},t)/\partial \hat{\mathbf{x}}$ in the RHS of Eq. (9) as a constant $\mathbf{C}$. In this way, the adjoint dynamics is relaxed as a linear ODE, the solution of which reads as follows:

$$\mathbf{a}(t) = \exp(-\mathbf{C}t). \tag{22}$$

Then we have the following relationship:

$$\mathbf{a}^2(n+1/2) = \mathbf{a}(n) \cdot \mathbf{a}(n+1). \tag{23}$$

As shown by Bhatia & Davis (1993), we have that the matrix form of the arithmetic-geometric mean inequality

$$2\|A^{1/2}B^{1/2}\| \le \|A+B\| \tag{24}$$

holds for any positive definite matrices $A, B \in \mathbb{R}^{n \times n}$ and unitarily invariant norm $\|\cdot\|$. We thus further bound above $\mathbf{a}(n+1/2)$ with $(\mathbf{a}(n)+\mathbf{a}(n+1))/2$ in Eq. (23). Substituting the upper bound for $\mathbf{a}(n+1/2)$ into Eq. (21) leads to Theorem 3.2. Notice that while our relaxation may have the potential negative effect on the accuracy of the splitting scheme itself, the formulation we provide differs from the LAD method and is easy to implement. Besides, slightly larger layer-wise perturbations may also contribute to the adversarial model robustness. We leave more accurate algorithms implementing the Strang-Marchuk splitting scheme for future work.

## B  EXPERIMENTAL SETTINGS

### B.1  GENERAL SETTINGS

Following He et al. (2016), we pad 4 pixels on each side of the image and sample a $32 \times 32$ crop from it or its horizontal flip. We use pre-activated ResNet-56 as our backbone architecture and experiment with our LAD (Eq. (5)) and LAD-SM (Eq. (8)) methods. For all experiments, we use the SGD optimizer with the batch size of 128. We train for 160 (300) epochs for the CIFAR-10 (-100) benchmark; the learning rate starts with 0.1, and is divided it by 10 at 80 (150) and 120 (225) epochs. We apply weight decay of 1e-4 and momentum of 0.9. We determine the scaling factor $\eta$ by cross-validation on training set, with $\eta = 50$ for CIFAR-10 and $\eta = 1000$ for CIFAR-100. For natural training, We set $\alpha = 0.5/255$ and $M = 20$ in the IFGSM attack. For PGD training, we set $\alpha = 2/255$, with iteration times $M = 10$ during PGD training and $M = 20$ for the IFGSM attacks. In the TRADES setting, we set $\lambda = 1/6$ and $\gamma = 1/2$. For other hyperparameters, we follow the settings in the PGD training experiments.

### B.2  IMPLEMENTATION DETAILS OF OUR METHODS

In implementation, we set hooks on each layers in the neural architectures and perform a forward pass with the perturbed inputs to calculate the loss. Gradients back-propagated from the loss function are caught by the hooks on each layers. In another forward pass, the recorded gradients are further scaled and added to each layer following Eq. (5) or Eq. (8). Following conventional adversarial training algorithms, we add the infinity norm bounded constraints to the input perturbations. We propose to replace the (adversarial) loss term with the lately calculated loss function. Denote that for a given data point $\langle \mathbf{x}, y \rangle$, the perturbed input is $\mathbf{x}'$. The natural loss term and the adversarial loss term are $L(f(\mathbf{x}), y)$ and $L(f(\mathbf{x}'), y)$, respectively. Then for natural training, we replace the term $L(f(\mathbf{x}), y)$ with the resulted loss function $L_{\mathrm{LAD/LAD-SM}}(f(\mathbf{x}'), y)$; for PGD training, we replace the term $L(f(\mathbf{x}'), y)$ with $L_{\mathrm{LAD/LAD-SM}}(f(\mathbf{x}'), y)$; for TRADES training, we replace the term $L(f(\mathbf{x}), y)$ with $\gamma L(f(\mathbf{x}), y) + (1-\gamma)L_{\mathrm{LAD/LAD-SM}}(f(\mathbf{x}'), y)$, where $\gamma$ is a hyperparameter for weighted averaging.

## C  STRONGER ATTACKS

Table 6 shows the accuracy and robustness results of our methods as well as baselines with adversarial training under stronger attacks on the CIFAR benchmarks. It can be seen that our methods have consistently outrun the baseline methods under attacks of different strength levels.

Table 6: Accuracy and robustness results of our methods as well as baselines under attacks of different strength levels on the CIFAR-10 and CIFAR-100 benchmarks.

| Benchmark | Method | Clean | IFGSM-20 | IFGSM-100 | IFGSM-1000 |
|---|---|---|---|---|---|
| CIFAR-10 | PGD | 82.14 | 44.79 | 44.25 | 44.18 |
| | PGD + LAD | 82.03 | 46.07 | 45.64 | 45.58 |
| | PGD + LAD-SM | 81.67 | 46.21 | 45.90 | 45.82 |
| | TRADES | 79.08 | 49.85 | 49.67 | 49.62 |
| | TRADES + LAD | 78.47 | 50.80 | 50.66 | 50.64 |
| | TRADES + LAD-SM | 78.49 | 50.68 | 50.57 | 50.54 |
| CIFAR-100 | PGD | 56.85 | 17.74 | 17.39 | 17.27 |
| | PGD + LAD | 58.11 | 22.44 | 22.29 | 22.26 |
| | PGD + LAD-SM | 57.97 | 22.39 | 22.21 | 22.17 |
| | TRADES | 52.53 | 25.86 | 25.80 | 25.78 |
| | TRADES + LAD | 52.46 | 27.10 | 27.05 | 27.05 |
| | TRADES + LAD-SM | 52.60 | 26.97 | 26.92 | 26.92 |

## D  RESULTS OF LAD WITH STOCHASTICITY ON CIFAR-100

In this section, we provide additional experimental results of LAD with stochasticity on CIFAR-100, shown in Table 7.

Table 7: Comparison of accuracy and robustness results of baselines and our methods with PGD training between the deterministic and the stochastic versions on CIFAR-100. The "Avg" column reports the averaged robustness results for each methods under different attacks of all the radii.

| Method | Acc | FGSM | | IFGSM | | Avg |
|---|---|---|---|---|---|---|
| | | 8/255 | 12/255 | 8/255 | 12/255 | |
| PGD | 56.85 | 23.71 | 16.10 | 17.74 | 8.52 | 16.52 |
| PGD + noise | 55.86 | 23.35 | 15.54 | 17.71 | 8.21 | 16.20 |
| PGD + LAD | **58.11** | 26.70 | 17.99 | 22.44 | 11.85 | **19.75** |
| PGD + LAD-SM | **57.97** | 26.79 | 17.94 | 22.39 | 11.58 | **19.68** |
| PGD + LAD + noise | **58.14** | 26.48 | 17.98 | 22.44 | 11.77 | **19.67** |
| PGD + LAD-SM + noise | **57.54** | 26.28 | 17.70 | 22.19 | 11.35 | **19.38** |

