# OpenReview forum: "Layer-wise Adversarial Defense: An ODE Perspective"
_ICLR.cc/2021/Conference — Reject_

### Official Review · AnonReviewer4 · 2020-10-27
**Interesting ODE perspective of layer-wise perturbations, inaccurate state-of-the-art and limited analysis**

**Rating:** 5
**Confidence:** 4

**Review:**

### Summary
The paper proposes to improve the robustness of residual networks (ResNet) by adversarial training with layer-wise and gradient-based perturbations. It further offers an interpretation of the proposed perturbations as ordinary differential equations with the lenses of the operator-splitting theory, resulting in two techniques (either using Lie-Trotter or Strang-Marchuk splits) for approximately computing the perturbations and training the model.

### Novelty and significance
Contrary to what stated in the paper, **layer-wise and gradient-based techniques have already been proposed** and analyzed in the past. For instance, layer-wise and gradient-based perturbations were first proposed in [1] and with a two stage approach, similar to the LT split approximation described here. Also in [2] the authors analyze the link between input perturbations and layer-wise perturbations.
**The real novelty of the paper is its interpretation of such perturbations on ResNet using ODE, and its link to split decompositions.**
This interpretation is interesting and significant as it provides theoretical foundation to the approximation schemes. However **its analysis in comparison with input perturbations falls short**:
1. it assumes that the model's output and the loss don't change much for small perturbations, which is not satisfied by common architectures, otherwise we wouldn't need to study adversarial examples and defenses;
2. it doesn't provide insights on why perturbing the intermediate representations improves robustness.

### Experiments
The paper carries out several experiments in the white box setting, where **the proposed techniques implemented with state-of-the-art adversarial attacks are shown to improve model robustness** overall and to perform better than methods that are tailored to  improve generic robustness.
The setup and results seem sound and the study of the perturbation scale $\eta$ is useful.
Of course, also in the experiments the proposed methods should be compared to existing gradient-based and layer-wise defenses that have been ignored.
Finally, it's not clear from the main text why the SM approximation works better than the LT one in practice.

### Quality and clarity
The paper is generally well written. It sometimes lacks of mathematical rigor and some important notations are missing:
1. In Problem (1) $L$ and $f$ are not defined and it should be clearly stated that the formulation is specific to ResNets.
2. In Equation (6) $\hat{x}$ is not defined.
3. $x$ and $\widetilde{x}$ are interchangeably used across the paper for original or perturbed inputs, which is confusing.

[1] Sankaranarayanan, Swami, et al. "Regularizing deep networks using efficient layerwise adversarial training." AAAI (2017)
[2] Chen, Xiaoyi and N. Zhang. “Layer-wise Adversarial Training Approach to Improve Adversarial Robustness.” IJCNN (2020).

---

### Official Review · AnonReviewer2 · 2020-10-29

**Rating:** 5
**Confidence:** 4

**Review:**

Summary

This paper proposes to extend the commonly used input perturbation in adversarial training to layer-wise perturbation, which adversarially perturbs all hidden layer inputs. It further analyses the perturbed dynamics from the perspective of ODE, and proposes two implementations based on different numerical schemes. These algorithms are empirically tested on CIFAR-10 and CIFAR-100 datasets.

I think this paper contains many interesting ideas, but they are not yet presented as a coherent story. In addition, the empirical results are not convincing enough to justify the authors claims. Therefore, I think the ideas here need further development before the paper can be accepted at ICLR.

Pros
1. This paper is overall clearly written, and provides enough background material on numerical analysis.
2. The ODE-based analysis is interesting, and this direction may provide further insight into related work.

Cons
1. A major problem is that the layer-wise perturbation is not well motivated and justified. Since adversarial attacks are only performed on the inputs, it is unclear to me why perturbing all hidden layers is necessary (deemed “essential” in multiple places in the paper) for the defense.  E.g., would hidden representations adapt from input perturbation through training?
2. The analysis, although interesting, is a bit handwavy. For example, why the last term in eq. 10 “can be negligible”? This claim needs to be justified by either proofs or numerical simulation that shows this term is indeed much smaller than others. In addition, what does the conclusion (“the main difference … lies in the first order initial values”) imply, and can it be experimentally verified?
3. The empirical results are not convincing enough. First, the reported numbers were averaged from 3 random seeds, while the deviations were not given. This is problematic given that the improvements from LAD or LAD-SM are relatively small -- without the standard deviation, it is difficult to evaluate their significance. In addition, as an extension of input perturbation, is LAD(-SM) comparable with PGD? It would be helpful to show columns cross Table 1, 2 and 3 with the same attack radii for such comparison.

Other comments

The comparison of different splitting schemes is interesting, but it is unclear why the Strang-Marchuk splitting scheme should be better in practice, since the ODE (eq. 7) is obtained from the limit of a discrete system. The experiments indeed show that LAD-SM was not superior in many cases. I would be helpful to discuss this.

---

### Official Review · AnonReviewer3 · 2020-10-30
**Interesting ODE interpretations of layer-wise adversarial training**

**Rating:** 5
**Confidence:** 3

**Review:**

The authors proposed a new adversarial training scheme based on ODE techniques. The standard adversarial training seeks for small perturbations in the input space. But the perturbations can be found in the feature space. From the observation and the similarity between the layer-wise adversarial attacks and ODE formulation, the optimization scheme was developed.

Clarity:
The paper is written well. Overall, it reads well. If the pseudocode of the two proposed methods is provided, it would be easier to understand the proposed schemes. The authors may save some space leaving out the proof of Theorem 3.3.

Strengths/Quality/Significance (pros):
The authors studied an interesting construction. The proposed methods show that the adversarial training can be generalized to hidden representations. The generalization can be viewed as ODE formulations.

To optimize the model parameters efficiently, using the operator-splitting theory from numerical analysis,
efficient numerical schemes can be developed using ODE theories. Discussion about Lie-Trotter Splitting Scheme and Strang-Marchuck splitting scheme was interesting.

In addition, the resulting algorithms achieved competitive performance compared to state-of-the-art methods.

The effectiveness of the proposed method with natural training methods is significant compared to vanilla natural training.

Weaknesses (cons) & Questions:
Authors claimed that  Eq 1. is inefficient to optimize and derive their numerical schemes based on techniques in the literature of ODE/Numerical analysis. The authors did not show the efficiency of the proposed methods. Even with a small dataset and small neural networks, if authors provide the comparison with Naïve approaches in terms of elapsed time/memory consumption and time/space complexity, it will be easier to evaluate the value of the proposed methods.

The effect of approximated schemes in terms of numerical stability and adversarial robustness needs to be discussed.

Since the bound for perturbation is omitted, the starting point (the optimization formulation) is unbounded. If authors tighten the loose end and bridge between the optimization problem and ODE-inspired numerical schemes, it will be more useful. Unlike the input space (image space), it is much trickier to define small perturbations to preserve the labels of samples. If the authors can fill this gap, it might be possible to provide a new definition of adversarial attacks.

It is confusing. In Figure 2, the smallest number (ratio) looks smaller than 10^5, unlike the discussion. Clarify this.

The performance gain against TRADES is table 3 is marginal.

No experimental results are available to verify the theoretical analysis of the second-order dynamics. Is it possible to design experiments to check whether the models/numerical schemes show the behavior?

---

### Official Review · AnonReviewer1 · 2020-10-30
**An interesting paper which needs major enhancement**

**Rating:** 4
**Confidence:** 4

**Review:**

This paper proposed a layer-wise adversarial defense which added perturbations in each hidden layer considering the influence of hidden features in latent space from the ODE perspective. It is essential to enhance the adversarial model robustness by stabilizing both inputs and hidden layers. The proposed method leveraged two operator splitting theory w.r.t. the Lie-Trotter and the Strang-Marchuk splitting schemes to discretize the specially designed ODE formulation by integrating the continuous limit of back-propagated gradients into the forward process. The main contribution of this paper is to generate perturbations with the idea of ODE in each layer. Empirical studies were performed to show the effectiveness of the proposed method on two benchmarks with two attack methods.

There are several concerns suggesting that this paper may not be accepted at its current form.
1.	Novelty is a concern.
a)	To improve the limitation that the existing adversarial training approaches mainly focus on perturbations to inputs, this paper added perturbations in each hidden layer with the ODE perspective, which is not convincing enough to me.
b)	The authors did not state the significance compared with  the other layer-wise perturbation methods. Why it is efficient to use the ODE method adding perturbation in each layer.
c)	At least, the authors should compare with the other adversarial training methods in a layer-wise way, e.g. the following two references.

Ref1. Sankaranarayanan, S.; Jain, A.; Chellappa, R.; and Lim, S. N. 2018. Regularizing deep networks using efficient layerwise adversarial training. In 32nd AAAI Conference on Artificial Intelligence.

Ref2. Kumari, Nupur, et al. "Harnessing the Vulnerability of Latent Layers in Adversarially Trained Models." IJCAI. 2019
2.	Another concern is that the paper may be in lack of sufficient and convincing experimentation to support the usefulness of the proposed method.
a)	The authors only selected one baseline method. As for the defense against attack method, this paper just used FGSM and IFGSM which are quite weak attacking methods. The proposed method may also be compared under strong attack methods, such as PGD and CW methods, since this paper targets defense.
b)	Furthermore, the results may not be significantly enough to verify the effectiveness of the proposed method.
c)	In Table1, did you use the same ResNet with the baseline, ResNet-110 or ResNet-164? It appears that the results in the paper are quite different from those reported in the original paper of Yang et al.
3.	The authors mentioned that they leveraged a two-stage approach to solve the time-consuming problem during training in section 3.1. There are not clear descriptions about this approach how it works and the authors did not explain why it could save time.
4.	In equation (1) and (5), some notations of x are different from each other. The authors did not give clear annotation which is confusing to me. The function f is not defined as well.
5.	In the experimental part, the authors did not conduct black-box experiments which are import in adversarial training.
6.	The paper could be further polished. There are quite a few typos. Figure 1 is not mentioned in the full paper. Table 4.1 should be Table 1 in the first paragraph in section 4.1.

---

### Decision · Program_Chairs · 2021-01-07
**Final Decision**

**Decision:**

Reject

**Comment:**

This submission aims to improve adversarial training by making it involve also layer-wise (instead of only input-wise) perturbations. This is an interesting idea and it is accompanied by an interesting ODE-based perspective on the resulting dynamics. However, as the comments and reviews detail, the current manuscript misses the discussion of very relevant previous work, does not specify important details of the approach (e.g., how to bound the extent of the perturbations used), and relies on weak primitives (FGSM vs PGD).

The consensus is that this would be an interesting and valuable contribution but only after addressing the above shortcomings.